# Hantavirus Replication Cycle—An Updated Structural Virology Perspective

**DOI:** 10.3390/v13081561

**Published:** 2021-08-06

**Authors:** Kristina Meier, Sigurdur R. Thorkelsson, Emmanuelle R. J. Quemin, Maria Rosenthal

**Affiliations:** 1Department of Virology, Bernhard Nocht Institute for Tropical Medicine, 20359 Hamburg, Germany; kristina.meier@bnitm.de; 2Centre for Structural Systems Biology, Leibniz Institute for Experimental Virology, University of Hamburg, 22607 Hamburg, Germany; sigurdur.thorkelsson@cssb-hamburg.de; 3Fraunhofer Institute for Translational Medicine and Pharmacology ITMP, 22525 Hamburg, Germany

**Keywords:** hantaviruses, structural virology, cryo-electron tomography, cryo-electron microscopy, X-ray crystallography, viral fusion glycoproteins, viral replication, viral transcription, viral genome encapsidation, virion assembly

## Abstract

Hantaviruses infect a wide range of hosts including insectivores and rodents and can also cause zoonotic infections in humans, which can lead to severe disease with possible fatal outcomes. Hantavirus outbreaks are usually linked to the population dynamics of the host animals and their habitats being in close proximity to humans, which is becoming increasingly important in a globalized world. Currently there is neither an approved vaccine nor a specific and effective antiviral treatment available for use in humans. Hantaviruses belong to the order *Bunyavirales* with a tri-segmented negative-sense RNA genome. They encode only five viral proteins and replicate and transcribe their genome in the cytoplasm of infected cells. However, many details of the viral amplification cycle are still unknown. In recent years, structural biology methods such as cryo-electron tomography, cryo-electron microscopy, and crystallography have contributed essentially to our understanding of virus entry by membrane fusion as well as genome encapsidation by the nucleoprotein. In this review, we provide an update on the hantavirus replication cycle with a special focus on structural virology aspects.

## 1. Introduction

*Bunyavirales* is a large viral order that includes many emerging viruses with high epidemic potential [1]. Notably, several bunyaviruses from the *Hantaviridae* family can infect humans, causing thrombocytopenia, capillary permeability (leading to vascular leakage), and immunopathology due to the activation of the innate and adaptive immune systems (reviewed in [2]). Whereas Old World hantaviruses, such as Hantaan virus (HTNV) and Puumala virus (PUUV), which are prevalent in Europe and Asia, can cause hemorrhagic fever with renal syndrome [3], the New World hantaviruses, such as Andes virus (ANDV) and Sin Nombre virus (SNV), which are found in the Americas, primarily cause hantavirus cardiopulmonary syndrome [4].

Humans are generally considered as dead-end hosts with the exception of some reports of person-to-person transmission for ANDV in Argentina [5], including recent cases of “super-spreaders” of the same viral strain [6] and the suspicion of transmission *via* the transfusion of platelets and blood products for PUUV in Finland [7]. Outbreaks of hantaviruses correlate with the population dynamics of their carriers, and each viral species seems to have a different and specific primary reservoir, which is usually either rodents or insectivore mammals such as moles, shrews, and bats [3,4]. Based on a rodent-hantavirus codivergence hypothesis, it has been proposed that hantaviruses have been co-evolving for a long time with their natural hosts, which are chronically infected with high viremia but remain asymptomatic [8]. However, the mechanisms that support hantavirus replication in a given host and that limit the spill over or adaptation to new organisms remain poorly understood (reviewed in [9,10]). Human infections upon exposure to contaminated aerosolized secreta or excreta have been reported almost exclusively from rodents so far, with case fatality rates ranging from 0.2% for the PUUV endemic in Europe [3] to ~50% in the case of ANDV in South America [4]. The possibility that hantaviruses circulating in insectivores can cause human diseases might have been overlooked and requires closer attention [2].

The enveloped virions are spherical or pleomorphic, are decorated with spikes, and encase the tri-segmented single-stranded viral RNA (vRNA) genome of negative polarity (Figure 1) [11,12,13,14]. The small (S), medium (M), and large (L) genomic segments encode four structural proteins: the nucleoprotein N, the glycoproteins Gn and Gc (resulting from the maturation of the glycoprotein precursor GPC after co-translational cleavage by the cellular signal peptidase complex), and the large (L) protein [15]. In some hantaviruses, the S segment also encodes a nonstructural protein (NSs) [16]. Although for ANDV, the NSs might work as an immunosuppressor of the type I interferon induction pathway [17], its functions in other hantaviruses and potentially additional roles during infection are unknown [18]. The envelope glycoproteins Gn and Gc form the spike complex responsible for receptor-binding and Gc-mediated membrane fusion. Integrins have been identified as receptors in vitro although there is no information on the receptors and co-receptors used in the natural context of infection (reviewed in [19]). Similar to other bunyaviruses such as the La Crosse virus (LACV, *Peribunyaviridae*) [20], the Crimean–Congo Hemorragic Fever virus (*Nairoviridae*) [21,22,23], and the Uukuniemi virus (*Phenuiviridae*) [24], hantaviruses appear to rely on several pathways for entry including macropinocytosis and endocytosis that is either clathrin-, calveolin- or cholesterol-dependent [19]. Particles then travel through the endocytosis pathway, and the low pH inside endosomes triggers a conformational change in the Gc glycoproteins [25]. This leads to the insertion of the Gc fusion loop into the endosomal membrane, the fusion of the latter with the viral envelope, and the subsequent release of the virion content. Each vRNA segment is flanked by non-coding regions at the 5′ and 3′ termini, exhibiting complementary sequences that are predicted to form a so-called panhandle structure, essential for the viral transcription and genome replication conducted by the viral L protein in the host cell cytoplasm. Based on recent structural data on the L protein, the formation of the panhandle might also be attributed to the L protein binding to both RNA ends [26,27,28,29,30,31]. Following the synthesis of virion components, the glycoproteins, which are specific for each hantavirus, play a key role in virus assembly and maturation (reviewed in [32]). In particular, the cytosolic tail of Gn is likely interacting with N proteins of the ribonucleoprotein complex [19]. Finally, it has been proposed that the virions being assembled bud into the Golgi apparatus and are released by exocytosis (Old World hantaviruses) or are released directly at the plasma membrane (New World hantaviruses), but the details of virion egress are largely unknown.

In this review, we summarize our current understanding of the molecular and structural biology of the hantavirus replication cycle, focusing on recent insights from structural virology studies. We discuss somewhat contradictory hypotheses regarding hantavirus entry, replication, and assembly. Finally, we highlight open questions in the field, which are not only critical to increase our knowledge on hantavirus–host interactions but to also develop specific countermeasures against emerging bunyaviruses.

## 2. Literature Review

### 2.1. Entry into the Host Cell

While bunyaviruses from the *Phenuiviridae* family have quite regular particles with T = 12 icosahedral quasi-symmetry and exhibit low plasticity in terms of size and shape [33,34,35], hantaviruses are pleomorphic with a diameter of 120–160 nm and vary from round to elongated [14,36]. The viral envelope is decorated with glycoproteins Gn and Gc in a grid-like pattern specific to hantaviruses [11,12,13,25]. However, this grid-like pattern can be also interrupted, resulting in bare patches of membrane on the virion surface [14]. Inside, the virion carries three genome segments (L, M, and S) which are encapsidated by the nucleoprotein N and are associated with the L protein (Figure 1) [15] (see assembly section).

Gc is a class II viral fusion protein with a three-domain architecture that is characterized by a β-sheet-rich secondary structure. Gn and Gc form the spike complex, which is by itself sufficient for host cell entry [13,37]. Consistent with biochemical data on the Tula virus (TULV), cryo-electron tomography analysis of HTNV, TULV, and PUUV have shown that the characteristic squared spike complexes are tetrameric and formed by the heterodimers of Gn and Gc (Figure 1 and Figure 2a) [12,13,38]. In the assembled fusion protein, Gc forms a lattice around the Gn protomers, which together form the spike through Gn:Gn interactions, maintaining Gc in a metastable pre-fusion conformation [25,37,38]. The fusion loop of hantavirus Gc is significantly different from related class II fusion proteins in that the residues involved in membrane insertion are split between three loops compared to all being on a single fusion loop for flaviviruses and alphaviruses (Figure 2b) [25,37]. The dissociation of the Gn/Gc complexes at a low pH is thought to make the tripartite fusion loop accessible for membrane fusion, although this dissociation does not seem to trigger membrane fusion and seems to even be reversible [39]. While Gc is responsible for membrane fusion, the role of Gn in viral entry remains unclear. However, it was recently shown that Gn can be a target for neutralizing antibodies against HTNV, emphasizing its importance for the spike function [40]. The conformational shift between pre- and post-fusion of Gn/Gc on viral particles is triggered by acidification in the endosome following virion uptake [34].

Although the cellular receptors in rodents are still unknown, there is evidence that the human integrins used are (i) ɑVβ3 in the case of SNV [41], HTNV, the Seoul virus, and PUUV [42,43]; (ii) ɑVβ1 for the Sangassou virus [44]; and (iii) β1 integrins for the Prospect Hill virus [41,42]. Apart from integrins, other cell surface proteins mediate virus entry in vitro, including the decay-accelerating factor CD55 and the complement receptor gC1qR/p32 [43,44,45,46,47,48]. So far, only protocadherin-1 has been shown to play a role in the entry of all New World hantaviruses (comprehensively reviewed by [19]). The hantaviruses ANDV and HTNV additionally depend on membrane cholesterol for host cell entry, most likely for endosomal escape and membrane fusion [49,50,51].

Despite the differences observed in the required macropinocytosis-related kinases ML-7 and ML-9 between HTNV, an Old World hantavirus, and ANDV, a New World hantavirus, both viruses enter human respiratory epithelial cells using a pathway that depends on sodium proton exchangers and actin, supporting that the entry process can involve micropinocytosis [52,53]. HTNV and PUUV also use clathrin-dependent endocytosis for host cell entry [53,54]. Additionally, it has been reported that HTNV, PUUV, and the Black Creek Canal virus (BCCV) tend to preferentially enter at the apical site of epithelial and endothelial cells [46,55]. However, more sound evidence is needed to uncover the intricacies of the hantavirus entry pathway(s). The current lack of relevant *in vitro* models is indeed a bottleneck to any study on hantavirus–host interactions in general, given their narrow host range and diversity [19,56].

After uptake, viral particles are transported to early or late endosomes [19], where the low pH of the compartment triggers large conformational changes in Gn/Gc and the Gn:Gc interface gets disrupted [40]. Similar to what has been observed for other class II fusion proteins [57], the transition of Gc from pre- to post-fusion is associated with rearrangements of the three domains in their relative orientation. First, a large relocation of the β-sheet stack at the C-terminus of domain III occurs. Upon additional reorganization of domain II, the buried key non-polar residues at the tip of the domain, constituting the tripartite fusion loop, become exposed on the molecular surface (Figure 2b). In fact, upon acidification, there is formation of a carboxylate–carboxylic acid hydrogen bond structuring the membrane-binding region [25,37,58]. Altogether, these conformational changes result in the assembly of protein monomers into trimers, which are stabilized by a conserved N-terminal segment or “N-tail”, which is so far unique to hantavirus Gc, along with membrane fusion, virion uncoating, and the release of the viral genome. Postfusion structures of class II fusion proteins are commonly trimers [57], which is also the case for the postfusion structures of ANDV Gc [37] and Gn/Gc [25]. The hantavirus-specific N-terminal tail of Gc was found to stabilize the Gc trimer [37]. Whether the three protomers of the fusion-active trimer originate from the same tetrameric prefusion complex or assemble from different tetrameric complexes during virion entry is still unclear.

### 2.2. Viral Genome Replication and Transcription

Viral ribonucleoproteins (vRNPs) consisting of the genomic RNA encapsidated by N protein and associated with the L protein are the functional units of genome replication and transcription (Figure 3). Both of these functions are catalyzed by the L protein that harbors the RNA-dependent RNA polymerase (RdRp).

Upon the release of the vRNPs into the cytoplasm, genome replication is initiated *de novo* and proceeds *via* a positive-sense complementary intermediate RNA (antigenome or cRNA) which is, similarly to the vRNA, encapsidated by N proteins. Interestingly, although self-initiating polymerases like the L protein typically use purines as initiating nucleotides, the hantavirus genome commences with a uridine monophosphate, suggesting initiation *via* a priming and realigning mechanism with the subsequent cleavage of the resulting overhang [59,60].

Although the bunyavirus vRNA and cRNA are always associated with N proteins, this is generally not the case for viral mRNA. However, it was reported that the encapsidation of the LACV (*Peribunyaviridae*) S segment mRNA with the N protein can occur, albeit with much lower affinity than that of vRNA and cRNA. Additionally, encapsidation of the S segment mRNA by the N protein was shown to prevent its own translation, which may serve as a negative feedback expression control at high cellular concentrations of N [61].

The bunyavirus genome segments are flanked by non-coding regions (also called untranslated regions, UTRs) which are highly conserved in each bunyavirus family. Based on the sequence complementarity of these UTRs, it was hypothesized that they promote circularization of the genome segments into a so-called panhandle conformation [62,63]. However, more recent structural data on closely related bunya- and orthomyxoviruses suggests that both RNA ends can be bound by the L protein at specific sites within the protein [26,27,28,29,30]. Thus, the genome circularization observed in electron microscopy (EM) (reviewed in [64]) might be the result of the L protein:RNA interaction rather than of panhandle formation *via* base-pairing but could also rely on both mechanisms (Figure 3b).

Severson et al. conducted binding studies with either (i) full-length vRNA; (ii) vRNA deletion mutants lacking either the UTRs or the first twelve 5′ terminal nucleotides; (iii) using oligonucleotides corresponding to the 3′; or (iv) 5′ termini of the vRNA only [65]. These studies led them to postulate a cis-acting encapsidation signal within the noncoding region of the HTNV 5′ terminal vRNA that would be recognized by the N protein. Upon specific recognition of this 5′ terminal signal, genome encapsidation would then be driven by the specific interactions among N protein monomers as well as proposedly sequence-unspecific interactions between N proteins and the remaining RNA along the genome or antigenome, respectively. While for HTNV, the N protein was reported to bind single-stranded RNA with a higher affinity than double-stranded RNA, the PUUV N protein has been reported to preferentially bind to double-stranded rather than single-stranded vRNA [66]. Along this line, SNV N was found to specifically bind to the panhandle composed of the 3′ and 5′ vRNA in an artificial minipanhandle RNA but not the single-stranded region of this RNA [67]. Whether the exact mechanism of RNA encapsidation by N is based on sequence-recognition and/or secondary structure-specific binding requires further investigation, including on the essential experimental controls, which were partially incomplete in the mentioned studies.

Crystal structures of the hantavirus N protein core revealed two lobes clamping a positively charged RNA binding groove with N- and C-terminal extensions linked to the respective lobe and mediating N protein oligomerization [68,69]. Recent cryo-EM studies showed that HTNV N forms a left-handed helical assembly, in which one N contacts 6 other protomers (Figure 3b). The N- and C-terminal arms of an N protomer interact with the N- and C-terminal lobes of the previous and successive protomers in the helix, respectively. The C-terminal extension can also rotate to contact the N-terminal arm of the same subunit, creating a positively charged groove in the helical assembly that is compatible with viral RNA binding [70]. Helix formation is likely coupled to RNA-binding since hantavirus N proteins were reported to be mostly trimeric [67] or hexameric [69] in solution. Indeed, Arragain et al. observed cellular RNA bound to the N protein after recombinant expression in the absence of viral RNA, suggesting that N protein:RNA interaction and the formation of the helical assembly may not depend on virus-specific RNA sequences [70]. It is still unclear how the L protein gains access to the encapsidated RNA, but interaction of the L protein and the N-terminus of the N protein seems to be necessary for viral RNA synthesis [71]. In the currently proposed model, this interaction with N brings the L protein in close proximity to the RNA, and the helical assembly of the N protomers would only be locally disrupted at the sites where the L protein reads the encapsidated RNA [70]. Of note, in addition to this compact helical conformation, a more flexible pearl-necklace-like conformation of the RNA-associated N protein has been reported [12,68], which would allow for more structural flexibility and may facilitate genome circularization.

Like other segmented negative-strand RNA viruses, hantaviruses employ a cap-snatching mechanism in which short, capped primers are cleaved off of host cell mRNAs and are subsequently used to prime viral mRNA synthesis [59]. The cap-snatching mechanism of the influenza virus (*Orthomyxoviridae*) has been extensively studied [72,73,74] and may occur analogously to that of bunyaviruses with the exception that orthomyxoviruses carry a heterotrimeric polymerase complex consisting of PA, PB1, and PB2 subunits. Additionally, orthomyxoviruses replicate and transcribe their genome within the cell nucleus, whereas bunyavirus genome replication and transcription occur in the cytoplasm [75,76].

The cap-snatching endonuclease of hantaviruses is located at the N-terminus of the L protein. It is classified as a His+ endonuclease, as it contains a catalytically important histidine residue upstream of the metal-coordinating PD-D/E-K active site motif. Overall, this domain is similar to the endonucleases of the influenza A virus PA subunit and the LACV L, but the hantavirus endonuclease has a significantly higher in vitro activity [77,78], which seems to limit its own recombinant expression in cells [79]. Sequestration of host cell mRNA has been postulated to occur at a distinct cap-binding site of the N protein, which would subsequently interact with the L protein for RNA cleavage and transcription initiation [80]. However, the structural characterization of the HTNV N protein did not yield evidence of a canonical cap-binding motif [69,70]. In contrast, recent studies provide structural and some functional evidence of a cap-binding domain in the C-terminus of arenavirus, phenuivirus, and peribunyavirus L proteins that are similar to the PB2 subunit of the influenza virus polymerase complex [27,30,81,82,83].

While the influenza polymerase complex has been shown to interact with the host cell polymerase II within the nucleus to gain access to mRNA caps [73,84], the specific cellular mRNA targets of bunyaviruses remain unclear (reviewed by [85]). Some studies suggest that hantaviruses snatch caps from non-sense RNA in processing bodies (P bodies), which are non-membranous cytoplasmic compartments specialized in RNA turnover, including the decapping machinery enzymes 1a, 1b, and 2 (DCP1a, DCP1b, DCP2) [86]. SNV has been reported to preferentially steal caps from mRNAs with premature stop codons [87], and SNV N was reported to colocalize with DCP1a [86]. Concomitantly, the transcription of the Rift Valley Fever virus (RVFV, *Phenuiviridae*) was shown to be restricted by the P body-associated decapping machinery, suggesting a shared pool of mRNA targets. Notably, RVFV N protein localization partially overlaps with P body-resident proteins [88]. In contrast, a recent study found no significant colocalization of TULV N protein and P body markers but reported significant colocalization of TULV RNA and N protein with stress granule-resident T-cell restricted intracellular antigen 1 (TIA-1) as well as an increase in the number of stress granules in TULV-infected cells [89]. Stress granules, similar to P bodies, are non-membranous compartments but rather serve as storage sites for translation-stalled mRNAs under cellular stress more than as sites of RNA decay. Stress granules contain small ribosomal subunits as well as translation initiation factors such as the eukaryotic initiation factors 4E and 4G (eIF4E, eIF4G). While eIF4E is also a constituent of the P bodies, eIF4G is restricted to stress granules [90,91]. However, stress granules and P bodies share some similarities in protein composition and are able to interact with one another and exchange RNA as well as possibly protein components (Figure 3a) [92,93,94]. Thus, both may serve as a site for cap-snatching and more research is required to further elucidate the roles of specific RNA granules during bunyavirus infection.

After transcription of viral genes *via* cap-snatching (Figure 3a), translation follows. For HTNV, it was proposed that the N protein could substitute the function of the eukaryotic initiation factor 4F complex (eIF4F), resulting in the preferential translation of viral mRNA [95,96]. However, structural characterization of HTNV N showed similarity to the tumor suppressor protein Programmed Cell Death 4 (PDCD4) [69], which disturbs the formation of eIF4F [97,98], and HTNV N may therefore also impede cellular mRNA translation by structurally mimicking PDCD4.

Another aspect of cytoplasmic replication is the possibility of coupling transcription to translation, which has been suggested for the Bunyamwera virus (BUNV, *Peribunyaviridae*) and may serve to avoid premature transcription termination [99].

Apart from the source of mRNA caps, the exact site of hantavirus genome replication and transcription has yet to be determined. Ramanathan et al. reported the formation of perinuclear structures by the HTNV N protein and their colocalization with markers of the endoplasmic reticulum–Golgi intermediate compartment (ERGIC) but only little colocalization with markers of only the endoplasmic reticulum (ER) or Golgi apparatus. Therefore, they suggested the formation of viral factories, virus-induced compartments, at the ERGIC and that the N protein would be quickly assembled into virions as soon as it reached the Golgi apparatus [100]. Davies et al. also reported perinuclear structures formed by the TULV N protein. These structures were of tubular nature and stained positive for Golgi markers and vRNA as well as for cRNA. Furthermore, they observed the recruitment of stress granules to these structures, suggesting the formation of viral factories within a structurally remodeled Golgi associated with stress granules (Figure 3c) [89].

### 2.3. Assembly and Egress of Viral Progeny

In the ER, the maturation of the glycoprotein precursor GPC by co-translational cleavage happens at the conserved pentapeptide motif WAASA [101]. The Gn and Gc glycoproteins then travel from the ER to the Golgi apparatus, and they oligomerize to form heterodimers [25,102,103,104,105]. Studies on PUUV revealed that Gn relies on Gc to be transported from the ER to the Golgi and that this process involves the C-terminal cytoplasmic tail of Gc [106]. In the full spike complex, the Gn/Gc dimer contains two glycan chains that are thought to originate from the ER, which stabilize the spike structure [25] and are essential for viral assembly and entry [103].

Although it is generally considered that Old World hantaviruses assemble in the Golgi apparatus like other bunyaviruses (e.g., the Uukuniemi Virus [107]), it has been proposed that the assembly of New World hantaviruses happens at the plasma membrane. This is based on the fact that the viral particles of the SNV and the BCCV were observed in the extracellular space close to the plasma membrane by conventional EM [55,108]. However, SNV virus-like particles were also found in the Golgi in one of these studies [108]. In addition, SNV glycoproteins could be detected at the plasma membrane at late infection time points [105]. However, SNV was detected in the perinuclear regions of the pulmonary endothelial cells in an EM pathogenesis investigation of tissue samples from patients [109] while analysis of a new hantavirus isolate in Vero cells reported several budding sites [110]. Recent work on the Old World HTNV and New World ANDV made use of high-pressure freezing and freeze-substitution to better preserve samples for EM and reported fragmentation and unstacking of the Golgi and herniation of the rough ER but no intracellular viral particles. At 7 and 9 days of infection, HTNV virions were observed extracellularly, close to projections of the plasma membrane, supposedly as a consequence of viral budding directly at the cell surface [111]. More systematic studies of both Old and New World hantaviruses, especially immunolabeling and 3D analysis of viral assembly and egress are still lacking to draw any conclusion on the assembly sites.

Gn/Gc octameric spikes serve a vital role in the budding of the virion. The lateral interactions between Gc proteins at the inter-spike interfaces have been proposed to be sufficient to induce the membrane curvature that would facilitate viral budding into the Golgi [13,38]. In fact, expression of the ANDV and PUUV Gn and Gc glycoproteins alone leads to the formation of virus-like particles released into the extracellular milieu [112]. In addition, there is significant reduction in virion production upon the disruption of the Gc:Gc inter-spike interfaces [25,38].

Gn and Gc are both type I integral transmembrane proteins with a C-terminal tail following the hydrophobic anchor domain [39]. The N protein was described to interact with both the cytoplasmic tail of Gc and Gn in PUUV [39]. Gn contains a 110 amino acid-long cytoplasmic tail, which interacts with N and might act as a substitute for the lack of matrix protein. This interaction is thought to be mediated by two zinc finger domains of the Gn cytoplasmic tail, which form a compact and unique fold [25,39,113,114]. Notably, the HTNV N protein also associates with RNA [67,70], forming ribonucleoprotein-like particles when expressed alone in mammalian cells but does not lead to the production of virus-like particles. For the recruitment of N into virus-like particles, the co-expression of Gn and Gc is required [115]. The N protein is essential for the encapsidation of the viral RNA segments by wrapping around the vRNA in a helical manner with 3.6 subunits per twist, fitting the negatively charged nucleotides into a positively charged groove within the helical structure [70] (see Section 2.2). *Via* interaction with the cytoplasmic tail of Gn, it confers recruitment of the genomic RNA segments into virions, but the exact sequence of these assembly steps is not yet clear. Some evidence suggest that the vRNA, the N protein, and the glycoproteins accumulate at the Golgi [89,116] and that microtubules are necessary for the correct movement of the viral components to the assembly site [100]. After budding into the Golgi apparatus, the viral particle is transported to the plasma membrane where it is then released *via* exocytosis. The involvement of multivesicular bodies or recycling endosomes in mature particle egress has been proposed and still remains largely unknown [117]. The N protein of the BCCV co-localizes with actin in infected cells, but the link between such association and viral assembly or release remains to be determined [118].

Of note, although the N protein can be detected in infected cells as early as 4 h post infection [100], putative viral factories were not perceptible at 36 h post infection, at which point the N protein was mostly localized in the perinuclear puncta. Larger, tubular structures of N protein assemblies became apparent at 7 days post infection, although no further time points between 36 h and 7 days post infection have been tested [89]. This is in accordance with reports noting the slow growth of hantaviruses [14], and hence, the assembly of N protein and other components into viral factories and recruitment of cellular structures as described in Section 2.2 may be a rate-determining step in the hantavirus replication cycle.

In summary, whether hantavirus assembly occurs at a specific site or at multiple locations remains unclear [55,108,109,110] as does the role of the cytoskeleton in viral component trafficking.

## 3. Discussion

The *Bunyavirales* order encompasses a very diverse number of viruses [1]. Members of the *Hantaviridae* family exhibit a number of specific features regarding their host range, also including insectivores and rodents, their direct transmission from persistently infected reservoirs to humans without using invertebrates as vectors, and their limited ability of human-to-human transmission, to name a few. Hantaviruses mostly grow very slowly and produce low titers in cell culture [119,120]. Although this slow viral growth rate seems rather inefficient, it might just be the reason that viral replication is possible at all, by staying below the radar of host’s antiviral defenses, but it makes working with these viruses more tedious. Additionally, there is a lack of relevant in vitro models [19,56], hampering studies on hantavirus–host interactions. Therefore, many details of the hantavirus replication cycle are still unknown.

First, the cellular receptor(s) used by hantaviruses to attach to cells as well as the interaction between the host receptor and the viral glycoprotein complex are not clear. Structural studies of viral particles by cryo-electron tomography and of isolated glycoproteins by X-ray crystallography greatly improved our mechanistic understanding of the low pH-induced conformational changes involved in membrane fusion [25,37]. Nevertheless, receptor-binding studies were unable to find a common determinant for hantavirus attachment and entry. Several candidates for receptors for attachment and entry have been described, but only protocadherin-1 could be shown to be relevant in all New World hantaviruses (comprehensively reviewed by [19]). Although hantaviruses are amplified in different cell lines, which cell types are initially infected in a natural infection has not been proven, which is in line with the lack of knowledge on the natural receptors used by hantaviruses. Second, a multitude of pathways have been proposed for the uptake of hantaviruses by the cell, including macropinocytosis and endocytosis that is either clathrin-, calveolin- or cholesterol-dependent [19]. Additional research is required to determine which of them is the road that is mainly used and whether this depends on the cell type, the exact virus species, or both.

Next, the mechanisms of viral replication and transcription also remain poorly understood. Recent structural data show how the hantavirus nucleoproteins can assemble to RNP-like structures in the absence of viral RNA with an impressive complexity of interactions observed between the N protomers and with a positively charged putative RNA binding groove buried inside the filamentous structure [70]. However, there is no structural data on the full-length L protein of hantaviruses, which contains the viral RdRp. The hantavirus L protein is a difficult target for in vitro expression due to its size of more than 200 kDa and the fact that its endonuclease domain is highly active, presumably digesting all sorts of RNA within cells, leading to a strong cytopathic effect, hence limiting its own expression [79] and possibly also general virus growth. Recombinant expression, biochemical characterization, and structure determination of the isolated endonuclease domain confirmed this hypothesis, as only significantly less active mutants of the domain could be produced, which were shown to digest a broad spectrum of substrates in vitro [77,78].

There are several structures of the full-length bunyavirus L proteins that have been published, such as for arenaviruses [30,121], phenuiviruses [28,83], and peribunyaviruses [26,27,31]. However, it remains unclear if these structures would enable reliable 3D modelling of the complete hantavirus L protein due to the mentioned diversity among bunyaviruses, which is also reflected in the L protein sequences and overall domain architecture. Previous attempts to predict the phenuivirus L protein structure based on the published structures of single domains of related polymerases did not yield entirely reliable models [122] when compared to experimentally determined structures [28,82,83]. However, this was before the game-changing appearance of AlphaFold, a novel structure-prediction tool with great potential [123].

While the endonuclease has been identified in the N-terminal domain of the full-length L protein [77,78], the cap-binding domain was hypothesized to be located in the N protein [80]. The latter is quite unlikely, as neither a canonical cap-binding site in the structure nor a convincing cap-binding activity have been demonstrated for N [69,80], which was reviewed by [85]. The current lack of structural information on the hantavirus L protein hence makes it impossible to propose a conclusive model of hantavirus cap-snatching. Recently, the expansion of the number of potential proteins expressed by bunyaviruses was proposed by a mechanism denoted as “start-snatching”. This process is based on cap-snatching and results from the introduction of a host start codon upstream of the viral open reading frames [124] but needs to be experimentally tested for hantaviruses.

Additional open questions relate to the exact cellular localization of viral genome replication and transcription as well as their regulation. Several studies present contradictory results regarding the replication and transcription site with the involvement of stress granules, P bodies, and different subcellular locations [86,89,100]. Importantly, most of the current studies used the overexpression of the N protein [71,86,100], which likely results in artificial distributions of the protein and the induction of cellular stress. As pointed out above, studies of infected cells are limited by the slow growth rate of hantaviruses and the low expression levels of the viral proteins [100,119,120]. However, further research and, in particular, research on the development of more relevant infection models is needed to answer these questions and to improve our understanding of the key stages of hantavirus infection.

After viral genome replication and transcription, the newly synthesized proteins and viral genomes have to be assembled to form new virions. Whether the genome segments are specifically recruited, as seen in influenza viruses [125], or packaged by statistical likelihood, as proposed for RVFV [126,127], is currently unknown. The cytoplasmic tail of Gc seems to be important for the recruitment of viral genomes into budding virions *via* an interaction with N [115]. Interestingly, the cytoplasmic tail of Gc (comprising 110 residues) has a size that is very similar to that of the Z protein specific to arenaviruses [128], which is not expressed by members of other bunyavirus families. Similar to the zinc-binding Z protein, the hantavirus Gc cytoplasmic tail was shown to contain a zinc-coordination site [113], and both have the same localization within the virion, lining the inner side of the viral envelope [128].

The cytoskeletal components important for transporting the subunits to the virion assembly site as well as the localization of the assembly site itself are also a matter of debate. The diverse scenarios propose assembly in the Golgi apparatus, the ERGIC, and the plasma membrane [89,100,105,109,110]. Similar to the studies of the N protein, investigation of the glycoprotein localization that is largely involved in the overexpression of the proteins and therefore potential artifacts have to be carefully considered. Whereas the sample preparation procedures for some of the earlier studies were also likely not optimal [55,108], a more recent study presented the same diversity of potential assembly sites [111], leaving it open as to if the assembly sites differ between hantaviruses or cell types or if they are simply dependent on the experimental setup (e.g., overexpression or sample preparation).

## 4. Conclusions

In summary, despite the enormous progress made in the field over the past years, many key aspects of the hantavirus replication cycle remain poorly understood. In particular, the mechanisms at play during genome replication, transcription, and particle assembly are unclear and are debated in the field. The biggest challenges seem to be the lack of suitable model systems and the slow viral growth rates in cell cultures, impeding detailed studies in the natural context of infection. We are confident that technological advances in the near future will aid in overcoming most of the noted barriers and will provide us with a comprehensive overview of hantavirus diversity and pathogenesis.

## Figures and Tables

**Figure 1 viruses-13-01561-f001:**
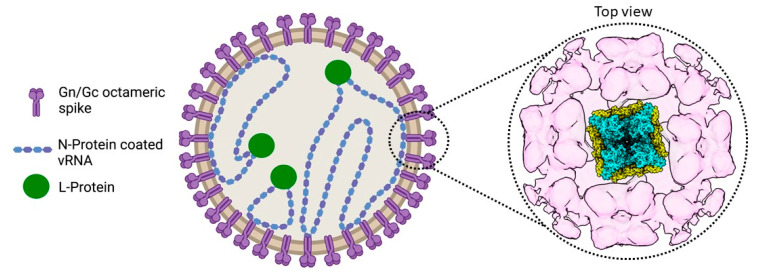
The hantavirus virion. A drawing of a hantavirus virion is presented (left) including a top view of the map of the TULV glycoprotein lattice (right), which was obtained by electron cryo-tomography and subvolume averaging (EMD-11236). A model of the prefusion tetrameric Gn/Gc spike complex (PDB: 6ZJM), presented as a cartoon, was fitted into the central volume [24].

**Figure 2 viruses-13-01561-f002:**
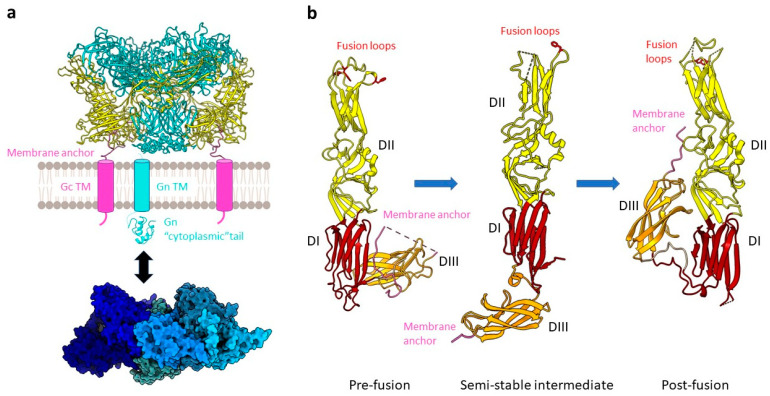
The hantavirus spike complex. (**a**) A pre-fusion spike complex on the virus surface composed of the Gn (teal) and Gc (yellow) proteins forming a tetrameric assembly (example of ANDV, PDB: 6ZJM) is presented as a cartoon. The viral envelope (grey membrane) as well as the transmembrane regions (TM) of Gn (cyan cylinder) and Gc (pink cylinder) are schematically shown. One copy of the Gn cytoplasmic tail (cyan, PDB: 2K9H) and a heptameric RNP-like assembly of the N protein (blue; PDB: 6I2N) are displayed as a cartoon and surface representation, respectively. An interaction between the latter is indicated (black arrow). (**b**) A comparison of pre-fusion ANDV Gc (PDB: 6Y5F), an intermediate state of HTNV stabilized by an antibody (antibody not shown) (PDB: 5LJY), and the ANDV Gc post-fusion conformation (PDB: 6Y6Q) is shown. Domains are colored as DI, red; DII, yellow; DIII, orange. Key residues W766, Y745, F900 for ANDV and F250 for HTNV of the tripartite fusion loop are highlighted in red with the side chains displayed as sticks.

**Figure 3 viruses-13-01561-f003:**
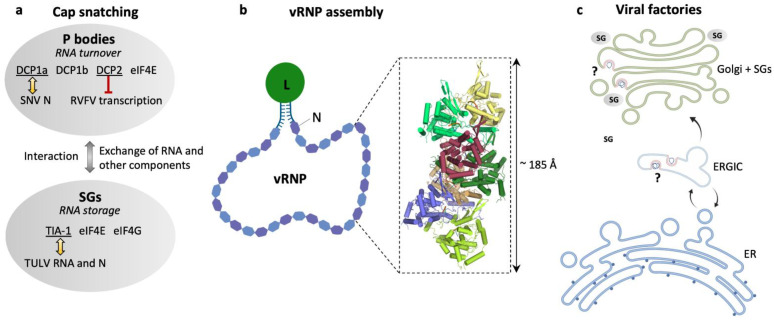
Hantavirus ribonucleoproteins and the location of viral genome replication and transcription. (**a**) Schematic representation of cellular RNA granules that have been reported as sites for hantavirus cap snatching: processing (P) bodies [86] and stress granules (SGs) [89]. Selected marker proteins for both granules are shown as well as eIF4E to emphasize its presence in both granules. Yellow arrows indicate reported colocalization of P body marker DCP1a and SNV N [86] or SGs marker TIA-1 and TULV RNA and N [89], and the red barred arrow indicates inhibition of RVFV transcription by P body-resident protein DCP2 [88]. (**b**) Schematic representation of a viral ribonucleoprotein (vRNP). The viral RNA is associated with N and L proteins in a panhandle-like conformation with the complementary genome ends forming a partially double-stranded region and the termini is most probably bound to the L protein. The close-up represents a heptameric assembly of N (PDB: 6I2N) into a helical RNP-like structure with tri-nucleotide RNAs (orange) bound to each N protomer. (**c**) Overview of possible locations of viral factories for genome transcription and replication in the endoplasmic reticulum (ER), endoplasmic reticulum–Golgi intermediate compartment (ERGIC) [100], and the Golgi apparatus associated with SGs [89].

## Data Availability

Not applicable.

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
