# Peer review of "Hantavirus Replication Cycle—An Updated Structural Virology Perspective"

_viruses, 2021, doi:10.3390/v13081561_

Round 1

Reviewer 1 Report

Reviewer comments - viruses-1285519-peer-review-v1

  • No major proof reading or English errors found. However, there are some minor errors. For example, line 19 in the abstract, Order “Bunyavirales” needs to be italicized per the naming conventions.
  • Literature referred to is exhaustive. However, authors should add one more paper when talking about morphology of any Hantavirus. Please refer to Parvate et al (2019,  https://pubmed.ncbi.nlm.nih.gov/31527500/). This is the most recent and most comprehensive report on morphology of purified hantaviruses. The author here has reported high resolution cryo-EM images of both OW and NW hantavirus morphology. Please make changes in the manuscript wherever morphology is mentioned and add to your citations. 
  • Section 2.1 – Organization of the GnGc spike – Would the authors want to make a comment on the most unusual tetrameric Gn-Gc spike? Is there any consensus regarding a fusion trimer formation i.e. In a GnGc tetramer, or a hetero-octamer - all 4 Gc spring into metastable post fusion conformation on exposure to low pH. But which 3 Gc actually form the fusion trimer? Is it 3 from the same GnGc tetramer or 1 each from 3 distinct neighboring GnGc tetramers? Or is this a random occurrence? Alternatively the authors can state this and allude to this being an open question.
  • Please make sure the paragraphs are well separated and are obvious. I am assuming the MDPI template may distort your original paragraphs.
  • Section 2.3 - Another incongruency overall in TEM studies of hantavirus infected cell lines is that - N protein is detected as early as 4 and as atleast by 12 hrs (Ramanathan et al, 2007-2008). But Virus particles take about 7 days to appear (Parvate et al 2020). Has any report provided any more data on this to explain why? Why have scientists waited upto even 14 days to actually see virus particles in infected cells? Would the authors want to comment on that?
  • Line 385 – Hantaviruses mostly grow

Author Response

Referee #1

No major proof reading or English errors found. However, there are some minor errors. For example, line 19 in the abstract, Order “Bunyavirales” needs to be italicized per the naming conventions. 

Answer: We thank the reviewer for pointing this out and carefully went through the manuscript to correct some remaining errors.

Literature referred to is exhaustive. However, authors should add one more paper when talking about morphology of any Hantavirus. Please refer to Parvate et al (2019,  https://pubmed.ncbi.nlm.nih.gov/31527500/). This is the most recent and most comprehensive report on morphology of purified hantaviruses. The author here has reported high resolution cryo-EM images of both OW and NW hantavirus morphology. Please make changes in the manuscript wherever morphology is mentioned and add to your citations.

Answer: We thank the reviewer for this suggestion, the mentioned manuscript is a valuable addition. The reference has been added to the text (lines 107 and 110) and the reference list has been updated accordingly.

Section 2.1 – Organization of the GnGc spike – Would the authors want to make a comment on the most unusual tetrameric Gn-Gc spike? Is there any consensus regarding a fusion trimer formation i.e. In a GnGc tetramer, or a hetero-octamer - all 4 Gc spring into metastable post fusion conformation on exposure to low pH.  But which 3 Gc actually form the fusion trimer? Is it 3 from the same GnGc tetramer or 1 each from 3 distinct neighboring GnGc tetramers? Or is this a random occurrence? Alternatively the authors can state this and allude to this being an open question.

Answer: This is indeed an interesting question, which hasn’t been solved yet. We have now included the following sentence “Postfusion structures of class II fusion proteins are commonly trimers [60], which is also the case for the postfusion structures of ANDV Gc [37] and Gn/Gc [25]. The hantavirus-specific N-terminal tail of Gc was found to stabilize the Gc trimer [37]. Whether the three protomers of the fusion-active trimer originate from the same tetrameric prefusion complex or assemble from different tetrameric complexes during virion entry is still unclear.” (lines 177-182).

Please make sure the paragraphs are well separated and are obvious. I am assuming the MDPI template may distort your original paragraphs.

Answer: We worked to improve this and hope that the current outline and separation of sections as well as paragraphs are easier to follow.

Section 2.3 - Another incongruency overall in TEM studies of hantavirus infected cell lines is that - N protein is detected as early as 4 and as at least by 12 hrs (Ramanathan et al, 2007-2008). But Virus particles take about 7 days to appear (Parvate et al 2020). Has any report provided any more data on this to explain why? Why have scientists waited upto even 14 days to actually see virus particles in infected cells? Would the authors want to comment on that?

Answer: We added a paragraph discussing this incongruency to section 2.3 (lines 380-388). To our knowledge it is unclear why hantaviruses have this extremely long replication cycle but this has been observed by many scientists and is very clear in the field even though not mentioned extensively in papers. The detectability of N early in infection is not entirely surprising as this is also the case for many other viruses and N proteins usually also inhibit the innate antiviral host defense. The late occurrence of viral particles points towards a somewhat inefficient process of viral genome replication, genome transcription and virion assembly. But maybe actually this slow process allows the virus to hide from the immune system and ultimately produce new viral particles. One fact that is stunning, is that the endoribonuclease located in the L protein and needed for cap-snatching is extremely active and unspecific, thereby limiting its own expression (Heinemann et al. 2013).

Line 385 – Hantaviruses mostly grow

Answer: the text has been edited according to the reviewer’s suggestion.

Reviewer 2 Report

In this manuscript, Meier et al. review the most relevant structural studies in the field of hantaviruses. I think it is well written and contains most of the relevant references. If the authors think is appropriate, here are some minor comments that I believe would improve their manuscript:

1. In the panel b of figure 2, the authors show a schematic representation of the extended conformation, they could instead use the Gc model of hantaan virus crystallised in complex with an antibody (5LJY) and which appears to take the extended form discussed by the authors. 

2. Lines 125-131. In this paragraph the authors discuss the cellular receptors used by hantaviruses, surprisingly they do not mention PCDH1, which is perhaps the best characterised receptor. They should include it. 

3. Lines 314-315. The authors mention that the Gn/Gc complex contains two glycan chains originated from the Golgi complex, but a number of studies have shown that hantavirus glycans are of the high-mannose type and are not modified in the Golgi apparatus. The authors need to clarify this point. 

4. Lines 350-351. The authors mention that "only upon co-expression of N, Gn 350 and Gc, virus-like particles are produced [113]", but different studies by N. Tischler's and T. Bowden's groups show that only Gn and Gc are necessary to form VLPs. The authors should discuss them.

5. Close parenthesis in lines 69, 84

6. For sake of clarity, in line 166 they should substitute "tail" by "N-tail"

Author Response

Referee #2

In this manuscript, Meier et al. review the most relevant structural studies in the field of hantaviruses. I think it is well written and contains most of the relevant references. If the authors think is appropriate, here are some minor comments that I believe would improve their manuscript:

  1. In the panel b of figure 2, the authors show a schematic representation of the extended conformation, they could instead use the Gc model of hantaan virus crystallised in complex with an antibody (5LJY) and which appears to take the extended form discussed by the authors.

Answer: We thank the reviewer for this suggestion and included the model 5LJY in figure 2.

  1. Lines 125-131. In this paragraph the authors discuss the cellular receptors used by hantaviruses, surprisingly they do not mention PCDH1, which is perhaps the best characterised receptor. They should include it.

Answer: We indeed forgot to include this information in section 2.1, we only mentioned this in the discussion (we noted this inconsistency just after submission). This important piece of information has now also been included lines 138-139 “So far, only protocadherin-1 has been shown to play a role in entry of all New World hantaviruses (comprehensively reviewed by [49]).”

  1. Lines 314-315. The authors mention that the Gn/Gc complex contains two glycan chains originated from the Golgi complex, but a number of studies have shown that hantavirus glycans are of the high-mannose type and are not modified in the Golgi apparatus. The authors need to clarify this point.

Answer: We thank the reviewer for their careful evaluation of our manuscript and are happy to be able to correct this mistake prior to publication.

  1. Lines 350-351. The authors mention that "only upon co-expression of N, Gn 350 and Gc, virus-like particles are produced [113]", but different studies by N. Tischler's and T. Bowden's groups show that only Gn and Gc are necessary to form VLPs. The authors should discuss them.

Answer: We are sorry for the confusion, this was exactly what we meant to say (as also noted in lines 354-355). We rephrased this sentence to improve clarity (line 363-367): “Notably, HTNV N protein also associates with RNA [70, 73], forming ribonucleoprotein-like particles when expressed alone in mammalian cells but does not lead to the production of virus-like particles. For recruitment of N into virus-like particles, co-expression of Gn and Gc is required [118].”

  1. Close parenthesis in lines 69, 84

Answer: Done.

  1. For sake of clarity, in line 166 they should substitute "tail" by "N-tail"

Answer: Thank you for the suggestion, the text has been edited (line 175).

Reviewer 3 Report

The mini review contains substantial information for the field. The paper is well organized and wrote by good English. I have no suggestion for the paper.

One minor error is the edge of Figure 3 out of range.

Author Response

Referee #3

The mini review contains substantial information for the field. The paper is well organized and wrote by good English. I have no suggestion for the paper.

One minor error is the edge of Figure 3 out of range.

Answer: We thank the reviewer for pointing this out. The figure has been resized to fit the template.
